Host-Microbe Biology

# A Family of Viral Satellites Manipulates Invading Virus Gene Expression and Can Affect Cholera Toxin Mobilization

Zachary K. Barth,[a] Zoe Netter,[a] Angus Angermeyer,[a] Pooja Bhardwaj,[b] [ID] Kimberley D. Seed[a,c]

[a]Department of Plant and Microbial Biology, University of California, Berkeley, Berkeley, California, USA
[b]Department of Laboratory Medicine, University of California, San Francisco, San Francisco, California, USA
[c]Chan Zuckerberg Biohub, San Francisco, California, USA

**ABSTRACT**    Many viruses possess temporally unfolding gene expression patterns aimed at subverting host defenses, commandeering host metabolism, and ultimately producing a large number of progeny virions. High-throughput omics tools, such as RNA sequencing (RNA-seq), have dramatically enhanced the resolution of expression patterns during infection. Less studied have been viral satellites, mobile genomes that parasitize viruses. By performing RNA-seq on infection time courses, we have obtained the first time-resolved transcriptomes for bacteriophage satellites during lytic infection. Specifically, we have acquired transcriptomes for the lytic *Vibrio cholerae* phage ICP1 and all five known variants of ICP1's parasite, the <u>p</u>hage inducible chromosomal island-<u>l</u>ike <u>e</u>lements (PLEs). PLEs rely on ICP1 for both DNA replication and mobilization and abolish production of ICP1 progeny in infected cells. We investigated PLEs' impact on ICP1 gene expression and found that PLEs did not broadly restrict or reduce ICP1 gene expression. A major exception occurred in ICP1's capsid morphogenesis operon, which was downregulated by each of the PLE variants. Surprisingly, PLEs were also found to alter the gene expression of CTXΦ, the integrative phage that encodes cholera toxin and is necessary for virulence of toxigenic *V. cholerae*. One PLE, PLE1, upregulated CTXΦ genes involved in replication and integration and boosted CTXΦ mobility following induction of the SOS response.

**IMPORTANCE**   Viral satellites are found in all domains of life and can have profound fitness effects on both the viruses they parasitize and the cells they reside in. In this study, we have acquired the first RNA sequencing (RNA-seq) transcriptomes of viral satellites outside plants, as well as the transcriptome of the phage ICP1, a predominant predator of pandemic *Vibrio cholerae*. Capsid downregulation, previously observed in an unrelated phage satellite, is conserved among <u>p</u>hage inducible chromosomal island-<u>l</u>ike <u>e</u>lements (PLEs), suggesting that viral satellites are under strong selective pressure to reduce the capsid expression of their larger host viruses. Despite conserved manipulation of capsid expression, PLEs exhibit divergent effects on CTXΦ transcription and mobility. Our results demonstrate that PLEs can influence both their hosts' resistance to phage and the mobility of virulence-encoding elements, suggesting that PLEs can play a substantial role in shaping *Vibrio cholerae* evolution.

**KEYWORDS** ICP1, PLE, RNA sequencing, *Vibrio cholerae*, bacteriophage, cholera toxin phage, viral satellite

V iruses are selfish genetic elements that reprogram their host cells for viral reproduction. Turning host cells into viral factories requires viruses to implement both their own tightly regulated gene expression programs and manipulations of host gene expression. Viral genomes can vary from just a couple of genes (1) to sizes rivaling those of cellular life (2), and so the gene expression strategies of viruses are highly varied.

Address correspondence to Kimberley D. Seed, kseed@berkeley.edu.

Viral satellites manipulate cholera phage gene expression

Viral lifecycles exist on a continuum of agency. Some, like the cholera toxin phage (CTXΦ), are relatively passive. CTXΦ exists as an integrated prophage within toxigenic *Vibrio cholerae*. CTXΦ is largely regulated by host stress and virulence regulons, producing cholera toxin during *V. cholerae* infection of mammalian hosts and replicating during the *V. cholerae* SOS response to DNA damage (3, 4). Aside from coding the two cholera toxin subunits, CTXΦ possesses a minimalist genome with just seven additional genes, five of which are structural or involved in virion morphogenesis. Upon induction, CTXΦ initiates its replication off the host chromosome. Assembled particles are released through host secretion machinery without killing the cell, allowing horizontal and vertical CTXΦ propagation (4). In contrast, many lytic phages have deadly mechanisms to shut down host gene expression and maximize expression of their own genes (5). These mechanisms often unfold in a concerted and controlled manner to give rise to tight temporal patterns of gene expression during infection, as has long been evidenced through targeted studies and more recently through global analyses such as RNA sequencing (RNA-seq) (6–13).

Less explored are the transcriptional patterns of viral satellites. These subviral elements parasitize viruses in a similar way to how viruses parasitize their host cells. Like viruses, viral satellites are found in all domains of life and impact their hosts in profound ways. Viral satellites can partially or completely abrogate virion production by the viruses they parasitize (14–16) and can reduce or worsen disease in multicellular organisms (17, 18). Unicellular organisms can be protected against viruses on the population level by endogenous viral satellites, but the efficacy of protection varies depending on the specific virus and satellite genotypes and infection context (15, 19–21). Given their broad distribution and importance for both their cellular and viral hosts, it is desirable to decipher how the reproductive programs of viral satellites intersect with and differ from the programs of the viruses they parasitize.

A prime model for mechanistic and evolutionary insights into viral satellites are the <u>p</u>hage inducible <u>c</u>hromosomal island-<u>l</u>ike <u>e</u>lements (PLEs) found in toxigenic *V. cholerae*. PLEs parasitize ICP1 (15), a lytic myophage that is the predominant phage in cholera patient stool samples (22). Following ICP1 infection, PLEs excise from the host chromosome and replicate to high copy number (15). Successful PLE parasitism does not abrogate cell lysis but results in the complete restriction of ICP1 and the release of PLE transducing particles (15) (Fig. 1). The tractable genetics of *V. cholerae* facilitates mechanistic studies of PLE gene products, and insights have been gained regarding chromosomal excision of PLE (23), PLE DNA replication (24, 25), and PLE manipulation of lysis kinetics during infection (26). Notably, ICP1 genome editing is also possible (27), allowing manipulation of both sides of this host-parasite relationship.

To date, five distinct PLEs have been identified within the genomes of *V. cholerae* isolates recovered from cholera patient stool samples dating back to the 1940s (15). Each individual PLE occurs in isolation; no *V. cholerae* isolate has been found to harbor more than one PLE, and PLEs typically dominate for a time before disappearing and being succeeded by a new PLE genotype (15). Four of the five known PLEs are integrated into repeats of the superintegron, an array of selfish and mobile elements in the *V. cholerae* small chromosome (15). PLE mobility, along with the extensive and growing library of ICP1 isolates, allows PLEs to be compared in shared strain backgrounds during infection by contemporaneous and noncontemporaneous ICP1 isolates (15, 23, 25). These experiments have shown that PLE and ICP1 are engaged in a coevolutionary arms race, with different pairings of PLEs and ICP1 isolates having different infection outcomes. The sole understood method through which ICP1 can overcome PLEs is the ICP1-encoded CRISPR-Cas system, and deletion of that system broadens the PLE and ICP1 interactions that can be studied (15, 20, 28). Pairing PLEs against the same host virus in an isogenic host cell background allows us to probe for convergence and divergence in how PLEs exploit and restrict ICP1. Thus, the ICP1-PLE system is a powerful model for exploring coadaptations between a virus and its satellite and is unparalleled for tracking how these adaptations have shaped the evolution of these warring elements.

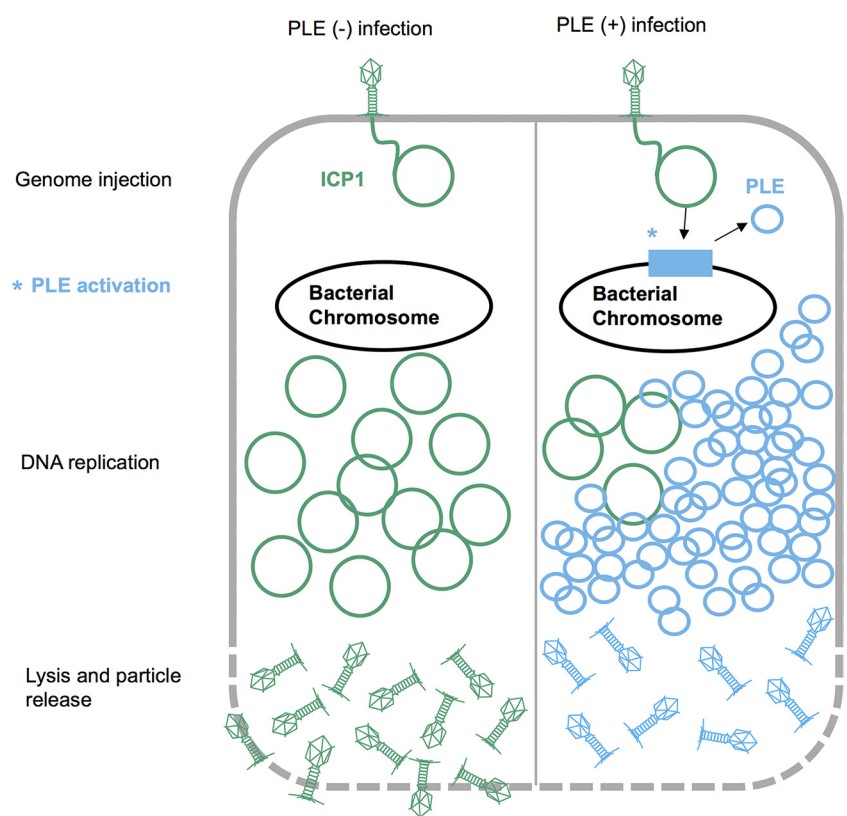

**FIG 1** PLE life cycle. Model of ICP1 infection in PLE(−) and PLE(+) *V. cholerae*. ICP1 injects its DNA into *V. cholerae*; prior to DNA replication, ICP1 activity leads to PLE activation and excision. ICP1 DNA replication is reduced in the PLE(+) cell, where the PLE replicates to high copy numbers. Finally, the cell lyses and releases infectious particles. Instead of ICP1 particles, PLE transducing particles are released from the PLE(+) cell.

So far, few insights have been gained into the gene expression programs of ICP1 and PLEs. PLE1 expresses its integrase in uninfected cells (23), expression of PLE1's replication initiator, RepA, is induced following infection of ICP1 (24), and the PLE's lysis modulator, LidI, is detectable by Western blot late during ICP1 infection (26). The PLE integrase's recombination directionality factor (necessary for directing integrase excision activity) is PexA, an ICP1 protein whose native function is unknown but whose expression can be detected by 5 min postinfection (23). While these limited observations have provided insight into key PLE and ICP1 genes, the gross expression patterns of ICP1 and PLEs remain unknown. Until now, we have not known the degree to which ICP1 alters cellular expression patterns and whether PLEs alter ICP1's gene expression or reproduce and restrict ICP1 without such alterations. To address these questions, we performed RNA-seq on *V. cholerae* infected by ICP1 over the course of the infection cycle. We sequenced the transcriptome of ICP1 infection in the absence of PLEs as well as in the presence of each of the five PLEs. This work deciphers ICP1's transcriptional program, the transcriptional program of each PLE, and the *V. cholerae* host transcriptome in each infection context. To our knowledge, this is the first detailed analysis of a viral satellite transcriptome during infection. Following ICP1 infection, PLEs exhibit remarkable conservation of temporal transcription patterns and targeted alteration of ICP1 transcription. The patterns described here suggest that like many viruses, viral satellites such as PLE have evolved to carefully coordinate gene expression. In contrast, when we compared uninfected PLE strains, we observed disparate interactions between PLEs and other mobile genetic elements in the *V. cholerae* genome. Surprisingly, most PLEs increase expression of the CTXΦ repressor *rstR*; however, the most recently circulating PLE, PLE1, upregulates CTXΦ's replication and integration factors, which we

show enhances the mobility of CTXΦ. Collectively, our findings show that successive PLEs have conserved interactions with ICP1 and divergent interactions with CTXΦ, providing insights into how satellites manipulate the gene expression of their host viruses and how they shape the evolution of their host cells.

## RESULTS AND DISCUSSION

*V. cholerae's* **response to ICP1 infection.** The ICP1 infection cycle takes approximately 20 min to produce a burst of nearly 90 infectious virions (15). To capture the temporal range of ICP1's infection cycle, we took samples for RNA sequencing immediately prior to infection and 4, 8, 12, and 16 min postinfection. Producing a large number of virions in a short period of time would presumably require substantial changes to the host cell transcriptome, and we see such changes occur during ICP1 infection. At 4 min postinfection, there are already dramatic changes to *V. cholerae's* transcriptome, with 17.2% (658/3827) of genes differentially regulated compared to that in uninfected cells (Q ≤ 0.1) (Fig. 2A and see Table S1, Tab 2 in the supplemental material). At this 4-min time point, slightly more host transcripts are predicted to be upregulated (345) than downregulated (313). When *V. cholerae* gene expression across infection is normalized to transcripts per kilobase million (TPM) and visualized by heat map, it appears that the bulk of *V. cholerae* genes are decreasing in transcript abundance following infection, and a small subset is being upregulated (Fig. 2B). We interpret the difference between our significant differential expression analysis results and TPM normalized expression profile to result from the differential expression analysis assuming a negative binomial distribution for gene expression changes (29). While such a model is appropriate for most RNA-seq applications where the majority of genes are not differentially expressed, it may underreport the extreme transcriptional changes that can occur during lytic viral infection. These extreme changes are reflected by the changes in read abundance over the course of infection. As had been noted previously, the *V. cholerae* transcriptome skews heavily toward genes on the large chromosome (30, 31). This imbalance persists throughout infection, but by 4 min postinfection, ICP1 contributes to more than one-quarter of total RNA reads within the culture, and by 16 min postinfection, ICP1 reads comprise more than 80% of total reads in the culture (Fig. 2C). These changes appear even more extreme when the relative size of the *V. cholerae* and ICP1 genomes are considered (Fig. 2D). Taken together, we interpret these data to show that ICP1 affects *V. cholerae* transcription by globally reducing *V. cholerae* gene expression while upregulating the expression of a subset of genes.

*V. cholerae's* most downregulated genes are enriched for tRNA and rRNA processing genes, while several different gene groups are enriched among upregulated transcripts (Fig. 2A and Table S1, Tab 2). The most dramatic differential expression of *V. cholerae* genes occurred in the ArgR regulon responsible for arginine biosynthesis. Arginine biosynthesis genes were highly upregulated upon ICP1 infection, with *argB*, *argC*, and *argF* expression increasing more than 100-fold at 4 min postinfection (Fig. 2A and Table S1, Tab 2). Similarly, the arginine transport genes were the most highly upregulated genes harbored by the *V. cholerae* small chromosome. The Na$^+$/H$^+$ antiporter encoded by *nhaA* was also strongly upregulated. Increases were also seen for genes relating to other amino acid biosynthesis and transport, sulfur compound metabolism, ATP production, flagellar synthesis and motility, and cell division (Table S1, Tab 2). In contrast to the decrease in tRNA and rRNA processing genes, we see an increase in ribosomal protein-coding genes and other genes involved in translation (Fig. 2A and Table S1, Tab 2). Several functional gene classes had many members upregulated and downregulated; for example, a large number of genes involved in transport were both up and downregulated, likely reflecting the difference in metabolic needs for lytic virus production versus that for normal cell growth. We also saw differential expression of cell envelope genes, with a decrease in some lipopolysaccharide biosynthesis genes and an increase in the mannose-sensitive hemagglutinin (MSHA) pilus associated with estuarine growth (Fig. 2A and Table S1, Tab 2) (32, 33).

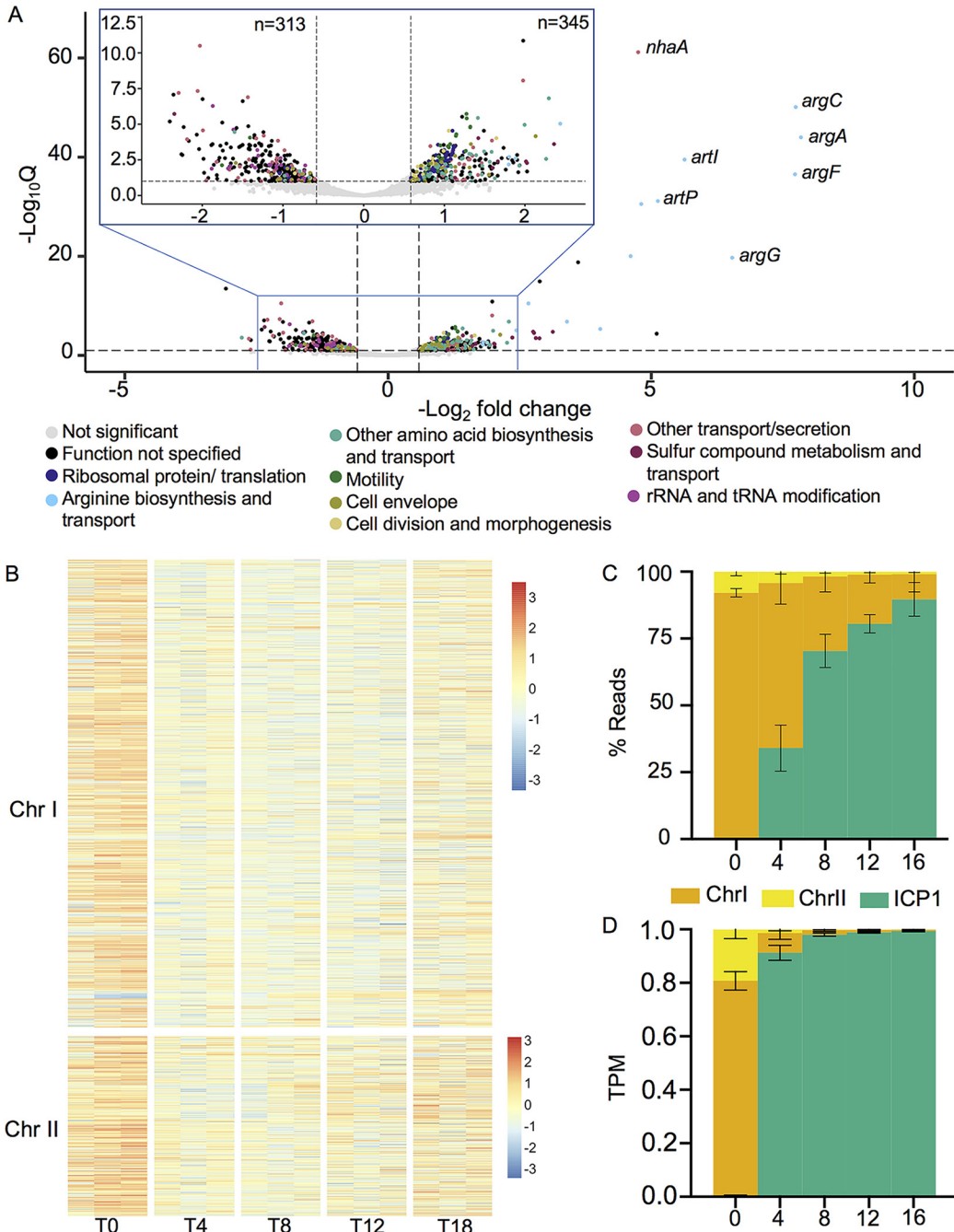

**FIG 2** The *V. cholerae* response to ICP1 infection. (A) Volcano plot showing gene expression changes between uninfected *V. cholerae* cultures and the same cultures 4 min postinfection. Full (bottom) and zoomed (top) views are provided to improve gene feature visibility. Genes are colored according to annotation. A significance cutoff of a Q value less than or equal to 0.1 and a $-\log_2$ fold change magnitude greater or equal to 0.585 (approximate to 1.5-fold) was used. (B) Heat map showing changes in *V. cholerae* gene feature $\log_2$ TPM over the course of ICP1 infection. Expression values obtained from three biological replicates are shown at each time point. ICP1 genes were excluded from TPM calculation to highlight relative changes in *V. cholerae* transcript abundance. Colors reflect the Z-score of each gene's $\log_2$ TPM value across replicates and time points. (C) Percent read abundances for both *V. cholerae* chromosomes and ICP1 over the infection time course. (D) Reads normalized to a TPM value based on the total number of reads from each element and the element's length. For all panels, results incorporate values obtained from three biological replicates.

It is hard to predict with certainty which transcriptional changes are a defensive host response to infection and which are due to transcriptional manipulation on the part of ICP1. Many of the differentially expressed genes relate to phenotypes regulated by cyclic di-GMP (di-cGMP) (34). Recently, another cyclic dinucleotide, cyclic GMP-AMP

(cGAMP), which was first discovered in *V. cholerae* (35), has been linked to phage defense (36). It is interesting to consider that ICP1 infection may be influencing cyclic dinucleotide signaling by triggering a host defense system; though in this case, the defense is not successful. Regardless of the source of these alterations, the expression changes we see are not fully consistent with a shift toward high or low levels of either cyclic dinucleotide. We see several gene changes consistent with high di-cGMP (34) (increased MSHA biosynthesis, increased cold shock, decreased heat shock, and increased type VI secretion expression) (Table S1, Tab 2) and some changes consistent with low di-cGMP (34) (increased flagellar synthesis and increased expression of the virulence regulator ToxT) (Table S1, Tab 2). An increase in MSHA is also consistent with low cGAMP; however, we also see downregulation of some chemotaxis-associated genes, and chemotaxis is repressed by high cGAMP (35). These changes suggest that the host cell may be receiving competing regulatory inputs that may act above or below the level of cyclic dinucleotide signaling.

The SOS response to DNA damage is often induced by phage infection, and several genes under SOS regulation have been identified in *V. cholerae* (37). We did not observe differential expression of genes under the SOS regulon following infection, suggesting that ICP1 may have a mechanism to avoid or repress this response.

The strongly increased expression of arginine metabolism and transport genes is especially curious and could have positive or negative effects on phage production. ICP1 may be upregulating arginine metabolism to drive the production of polyamines or purines. Arginine often serves as the precursor for polyamine synthesis (38). Purines may be a limiting resource for phage genome replication, while polyamines are found in the capsids of multiple phages, potentially aiding in DNA condensation, packaging, and ejection (39, 40). We do not see upregulation of genes specific for synthesis of the most common polyamine, putrescine, but the ornithine decarboxylase which is responsible for putrescine synthesis is regulated posttranslationally in *Escherichia coli* and mammals to allow rapid adjustment of polyamine pools (38, 41). We also see upregulation of the *V. cholerae*-inducible lysine decarboxylase (Table S1, Tab 2), and in at least one bacterial system, a lysine decarboxylase is able to use ornithine as a substrate (42). Alternatively, arginine has been found to have a deactivating effect on phage virions under certain conditions (43–45), and so the upregulation of arginine may be an attempt by the host cell to curb phage production.

**Establishing the ICP1 transcriptional program.** Once we examined ICP1's effect on *V. cholerae* transcription, we next sought to document ICP1's transcriptional program. ICP1 has an approximately 126-kb genome and more than 200 predicted open reading frames (46). Less than one-quarter of ICP1's putative coding sequences have activities or functions that can be predicted through bioinformatic analysis. As is common for viruses, ICP1's genes fall into distinct temporal groupings based on the timing of peak expression. The early genes, which we define as those that show peak expression at 4 min postinfection, consist mostly of short genes averaging less than 330 bp and encoding hypothetical proteins (Fig. 3 and Table S1, Tab 3). It is difficult to infer the function of these genes, but there are several short immediate early genes in other phage systems that are known to have a role in host cell takeover (47, 48). The next grouping, middle early genes with peak expression at 8 min postinfection, mostly comprise nucleotide metabolism genes, including ICP1's DNA polymerase (Fig. 3 and Table S1, Tab 3). The 12-min time point captures a transitional period in ICP1 transcription. High expression of the middle early genes continues, while expression of late genes has begun but not yet peaked (Fig. 3). Few genes hit peak expression at 12 min, though an exception is a subset of nucleotide metabolism genes (within the range *gp176* to *gp211*). Finally, late genes with peak expression at 16 min postinfection comprise primarily putative structural genes and genes known to be involved in lysis (Fig. 3 and Table S1, Tab 3) (26). The lysis, capsid, and tail genes occur in three separate clusters, all encoded on the (−) strand (Fig. 3 and Table S1, Tab 3). Previously, ICP1 rolling circle replication was predicted to proceed in the (−) direction based on late

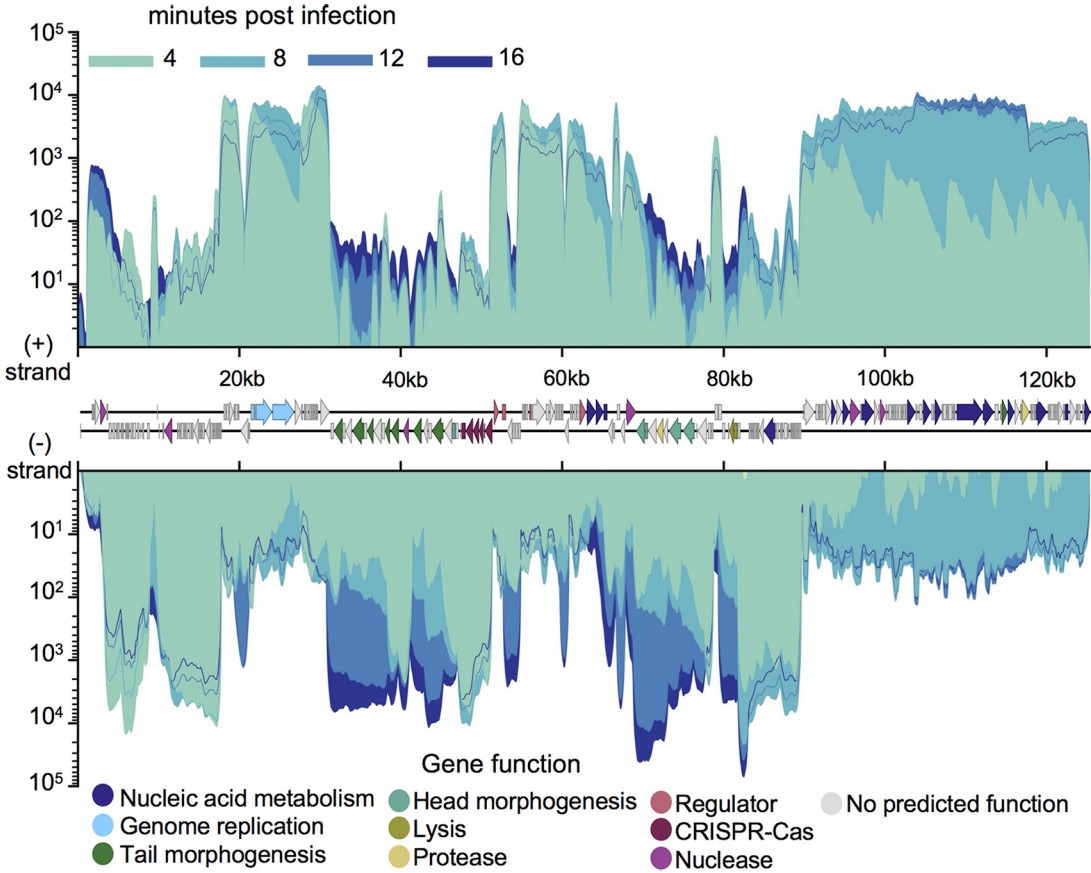

**FIG 3** ICP1 transcriptional patterns. ICP1's genome displaying average read coverage over the course of infection of PLE(−) *V. cholerae*. Reads are color coded by time point. Gene features are colored based on known or putative gene functions. Results incorporate values obtained from three biological replicates.

infection DNA coverage skews (24). Since ICP1's late genes are transcribed off the (−) strand, they may be transcribed off the rolling circle replication template. Such an arrangement would be consistent with the preference for codirectional transcription and DNA leading strand synthesis observed in many bacterial systems (49) and is thought to help preserve genome integrity by avoiding replication and transcription conflicts (50). Overall, the expression patterns we see for ICP1 are consistent with what is known about lytic phage development in general and ICP1's life cycle in particular.

**PLEs exhibit a conserved transcriptional program.** Having established the transcriptional program of ICP1, we next sought to examine transcriptional patterns in PLEs. The five PLEs share a similar gene organization. Proximal to the PLE's left attachment site is a gene cluster (denoted $C_{int}$) that includes the PLE integrase, *int*. Immediately downstream of this gene cluster is PLE's sole cluster of negative-strand genes, here called $C_L$ for "left" cluster (Fig. 4A). This (−) sense cluster flanks an approximately 3-kb noncoding region, the right most one-quarter of which contains the PLE origin of replication (24). Flanking the PLE origin of replication is a putative *marR*-like gene, and this is followed by two additional rightward facing clusters ($C_{R1}$ and $C_{R2}$). Exceptions to this arrangement occur with PLE1's standalone *int*, a transposase in PLE2's large noncoding region, and the absence of a *marR*-like gene in PLE3 (15).

Consistent with the PLEs providing interference against this ICP1 isolate (15), we find that all PLEs are transcriptionally activated following infection. Paralleling PLEs' organizational similarities, the transcriptional patterns of PLEs are highly conserved once each PLE has been activated. In uninfected samples, PLEs show some expression of *int*, and, if it is present, the *marR*-like gene, while expression across $C_L$, $C_{R1}$, and $C_{R2}$ is variable and often uneven (Fig. 4B and Fig. S1). Expression of *int* in uninfected cells is

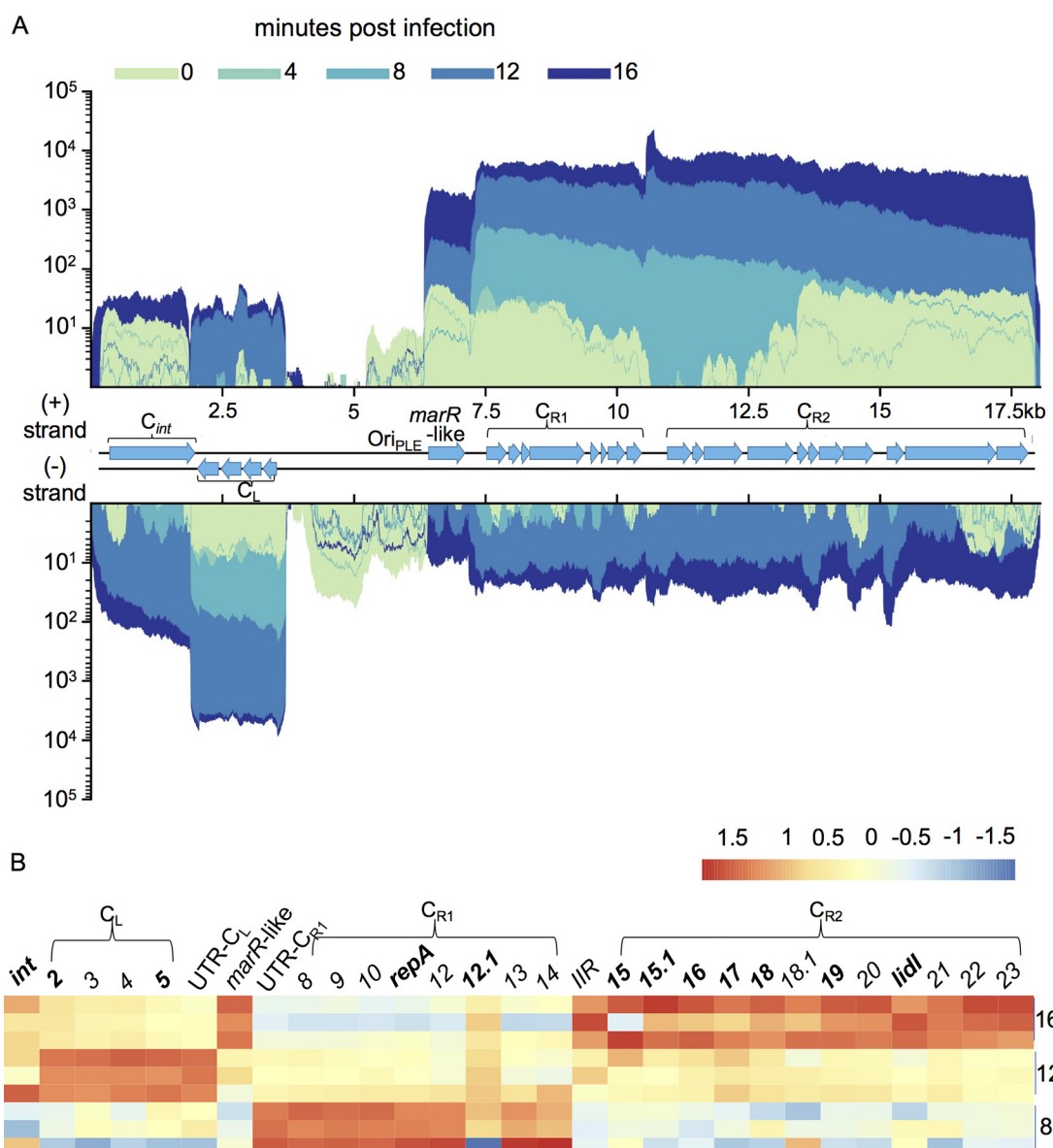

**FIG 4** PLE transcriptome. (A) PLE1's genome displaying average read coverage over the course of infection. Reads are depicted on a logarithmic scale to improve visibility of early expressed genes. Reads are color coded by time point. (B) Heat map of PLE1 gene expression over the course of infection. Color reflects the Z-score of each gene's $\log_2$ TPM value across replicates and time points. *V. cholerae* and ICP1 genes were excluded from TPM calculation to highlight relative changes in PLE1 transcript abundance. Core genes, protein-coding genes with high conservation across PLEs, are bolded. Values for 8, 12, and 16 min postinfection are shown. Results incorporate gene expression values obtained from three biological replicates.

consistent with previous work, where Int was detectable by Western blot in the absence of ICP1 (23). Read counts for PLE genes remain low at 4 min postinfection (see Data Set S1 Data), and then at 8 min postinfection, transcriptional patterns start to emerge. For each PLE, $C_{R1}$ is strongly expressed at 8 min postinfection, followed by strong expression of $C_L$ at 12 min, and high $C_{R2}$ expression at 16 min along with the *marR*-like gene (Fig. 4 and Fig. S1). It should be noted that all PLE transcripts continue to increase in abundance over the course of infection, though we see differences in the timing of peak gene transcription relative to that of other genes. This sustained global increase can likely be attributed to increased PLE copy number, as PLE replicates upwards of 1,000-fold during ICP1 infection (15).

Early expression of $C_{R1}$ is not surprising, given that this cluster contains PLE's *repA* gene, which is necessary for PLE replication (24), and PLE1 replication was previously

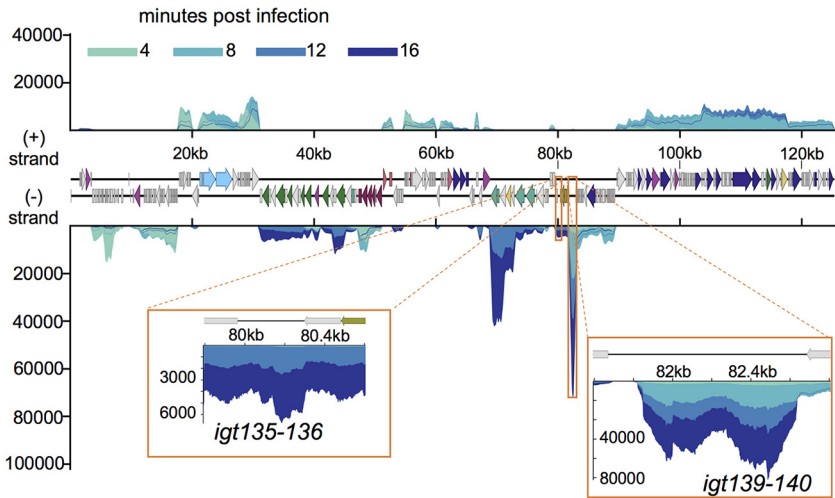

**FIG 5** ICP1 noncoding RNA. ICP1's genome displaying average read coverage over the course of infection on a linear scale. Reads are color coded by time point. ICP1 gene features are colored based on known or putative gene functions as in Fig. 3. Regions boxed in orange show transcripts that lack predicted coding sequence. Results incorporate gene expression values obtained from three biological replicates.

found to begin before 15 min postinfection (15). Interestingly, a highly conserved $C_{R1}$ gene, *orf12.1* in PLE1, has a different expression pattern than the rest of the cluster, peaking at 12 min postinfection instead of 8. This suggests that *orf12.1* may be under different regulation than the rest of $C_{R1}$ and may be involved in the transition from early to late PLE gene expression. Overall, the conserved timing of expression of syntenic gene clusters across PLEs suggest that each cluster serves a distinct role in parasitizing ICP1. We suspect that the timing of PLE gene cluster expression has evolved to take advantage of ICP1's own transcriptional program. Coordination between PLE gene expression and ICP1's gene expression would be consistent with PLEs' reliance on ICP1 gene products for key steps of the PLE life cycle (23–26).

**Noncoding RNAs are abundant in ICP1 and PLEs.** Noncoding RNAs (ncRNAs) are a prominent feature of both ICP1 and PLE gene transcription. Surprisingly, the most abundant transcripts in the ICP1 and PLE transcriptomes are both predicted to be noncoding. The most abundantly expressed ICP1 transcript is in an approximately 1-kb orf-less space between *gp139*, the start of ICP1's lysis cluster (26), and *gp140* (Fig. 5). The length of this transcript (~800 bp based on the RNA-seq coverage), is comparable to that of the giant-, ornate-, lake-, and *Lactobacillales*-derived (GOLLD) and the rumen-originating, ornate, large (ROOL) RNAs that have been found in many phage genomes (51). These RNAs are frequently harbored near tRNAs, although this is not the case for ICP1, since it does not possess any tRNAs. The role of these large ncRNAs is unknown. In one *Lactobacillus brevis* prophage, a GOLLD RNA was found to accumulate during lytic infection but was dispensable for phage production (51).

In each PLE, the most abundant transcript is located between $C_{R1}$ and $C_{R2}$ (Fig. 6 and Fig. S2). This transcript occurs between a set of inverted repeats, and for this reason, we have tentatively named the transcript the interinverted repeat (IIR) transcript (see Fig. S3). The PLE IIR transcript also has antisense homology to the leader sequences of several PLE open reading frames (ORFs) (Fig. S3B). Complementarity between an ncRNA and gene leader sequences is seen in the regulatory RNAs from phages P1, P7, and N15 as well as the phage satellites P4 and ΦR73 (52–54), suggesting that the IIR transcript may have a role in regulating PLE gene expression. Noncoding RNAs flanked by terminal inverted repeats often occur in miniature inverted-repeat transposable elements (MITEs) (55), suggesting that the PLE IIR transcript and the aforementioned phage RNAs may have been "domesticated" from a class of mobile genetic elements.

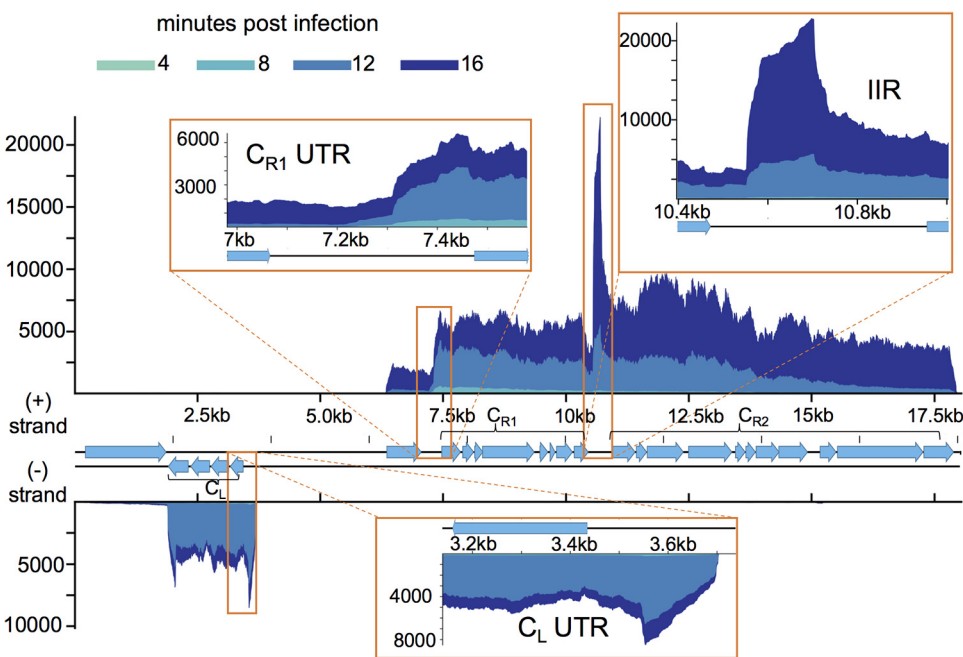

**FIG 6** PLE1 noncoding RNA. PLE1's genome displaying average read coverage over the course of infection on a linear scale. Reads are color coded by time point. Inserts depict detected transcripts that lack predicted coding sequence. Results incorporate gene expression values obtained from three biological replicates.

Additional transcripts without predicted coding capacity occur in both ICP1 and PLEs. An approximately 300-bp region between *gp135* and *gp136* is transcribed in ICP1 between 12 and 16 min postinfection (Fig. 5). Within each PLE, we see abundant transcription approximately 200 bp upstream of $C_L$ and 150 bp upstream of $C_{R1}$ when these clusters are transcriptionally active (Fig. 6 and Fig. S2). Though these 5' untranslated regions (UTRs) are not conserved on the sequence level, their occurrence in every PLE suggests conservation of function. The 5' UTR of transcripts is a common site for riboswitches (56), suggesting that these untranslated sequences may regulate expression of their downstream genes.

**PLE-host interactions in the uninfected cell.** By comparing transcriptomes of PLE(+) and PLE(−) strains prior to infection, we were able to determine whether PLEs affected transcription in their *V. cholerae* host prior to phage infection. All PLEs altered the transcription of genes neighboring the PLEs' integration sites (Fig. 7A and Fig. S4). Additionally, PLEs altered the expression of several genes within the *V. cholerae* superintegron, including multiple toxin-antitoxin systems (Table S1, Tab 4). Altered expression of superintegron genes also occurred in the PLE2(+) strain, despite PLE2 being integrated outside the superintegron (15). These transcriptional changes may reflect cross talk between PLE-harbored genes or genes flanking the PLE integration site and other genes within the superintegron. Notably, many of the superintegron genes are multicopy, and so the number of differentially regulated genes may be overreported if the read mapping cannot differentiate reads from multicopy genes from different loci. This explanation seems particularly likely for scenarios where genes that are distal to the PLE integration site are predicted to be differentially expressed and have paralogs located proximal to the integration site. Such a pattern is seen with PLEs 3 and 5 (Table S1, Tab 4) but can only explain a small number of the differences we see in the superintegron.

Rather surprisingly, we also found that PLEs alter the expression of genes harbored by integrative mobile elements exploiting Xer (IMEXs) (Fig. S4). IMEXs are mobile elements that utilize host Xer recombinases to integrate into the chromosome dimer resolution or *dif* sites of their host cells, located near the chromosome replication

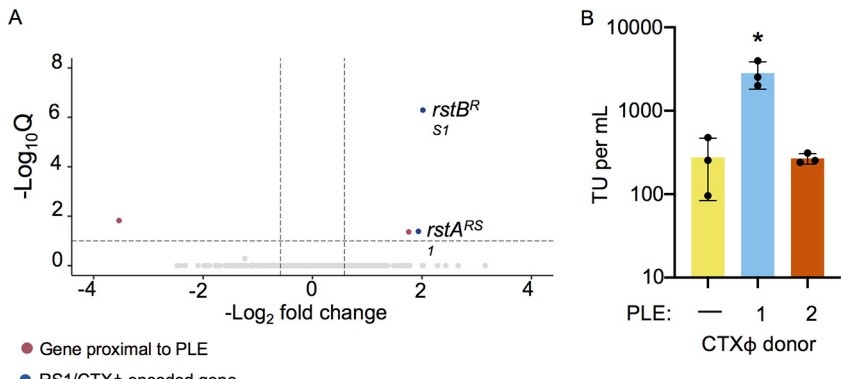

**FIG 7** CTX production is upregulated in PLE1-positive strains. (A) Volcano plot showing differential regulation of *V. cholerae* genes in uninfected PLE1(+) cultures relative to that in PLE (−) cultures. Genes within 2 kb of the PLE integration site are colored red, and genes harbored by RS1 and CTXΦ are colored blue. Genes that are not significantly differentially regulated are colored light gray. A cutoff of a Q value less than or equal to 0.1 and a −log$_2$ fold change magnitude greater or equal to 0.585 (approximate to 1.5-fold) was used. (B) The transduction units per milliliter of CTXΦ produced by PLE(−), PLE1(+), and PLE2(+) cultures following induction by mitomycin C. Statistical significance was determined using a one-way analysis of variance (ANOVA). *, $P = 0.04$.

terminus (57). The strain of *V. cholerae* used in this study has three separate IMEX's: CTXΦ, CTXΦ's satellite RS1Φ, and the toxin-linked cryptic element (TLC), integrated twice in tandem next to CTXΦ (Fig. S4E). RS1Φ is largely redundant with a sequence within CTXΦ. RS1Φ and CTXΦ have their own copies of the replication initiator *rstA*, a gene required for integration named *rstB*, and the CTXΦ master repressor *rstR* (3, 4). All these IMEXs are integrated on chromosome I, and so the PLEs' transcriptional effects on these elements must be acting in *trans*. Relative to that in the PLE(−) strain, most PLEs increased expression of both the RS1Φ and CTXΦ copies of *rstR*. These same strains also showed upregulation of the TLC gene *tlcR*. An exception to this pattern occurred in PLE1, where rather than *rstR* upregulation, we observed upregulation of the CTXΦ replication genes *rstA* and *rstB* (Fig. 7A). This observation prompted us to question whether upregulation of *rstA* and *rstB* could prime CTXΦ for mobilization following activation of *V. cholerae*'s SOS response. To test this, we used an antibiotic marked copy of CTXΦ and found that following mitomycin C treatment, the presence of PLE1 increased the production of CTXΦ-transducing units 10-fold relative to that in a strain with no PLE or a strain with PLE2 (Fig. 7B). These results reveal potentially far-reaching effects that mobile elements can have on each other as well as the hosts they share. Beyond providing the host cell population with immunity to ICP1 phages, PLEs' integration can enhance the mobility of other mobile genetic elements and, by extension, virulence genes. Thus, PLEs may affect *V. cholerae* fitness in ways that are distinct from their own antiphage activity and relevant to cholera epidemiology.

**PLEs selectively manipulate ICP1 transcription.** Having detailed ICP1's and PLEs' transcriptional programs and PLEs' transcriptional effects in uninfected cells, we sought to evaluate whether PLE disrupted ICP1 gene expression. Remarkably, although PLEs abolish ICP1 production (15), we found that PLEs do not broadly restrict or alter ICP1 transcription. At 16 min postinfection during maximum PLE expression, PLE transcripts comprise roughly 10% of the transcriptome, while ICP1's proportion of reads still sits at around 80% (Fig. 8A). When normalized to the genome size, PLE1 TPM approaches parity with that of ICP1, while the other PLEs achieve a bit less (Fig. 8B and Fig. S5), indicating that the relative transcriptional activity of PLEs does not exceed that of ICP1. This is in stark contrast to previously reported DNA levels at 16 min postinfection, where the amount of PLE1 DNA exceeds that of ICP1 and overall ICP1 DNA replication is substantially reduced by PLE1 (24). During ICP1 infection of PLE1-positive cells, there is a loss of ICP1's rolling circle replication (24). PLE1's disparate effects on ICP1 replication and transcription can be reconciled by the model that ICP1 late genes are

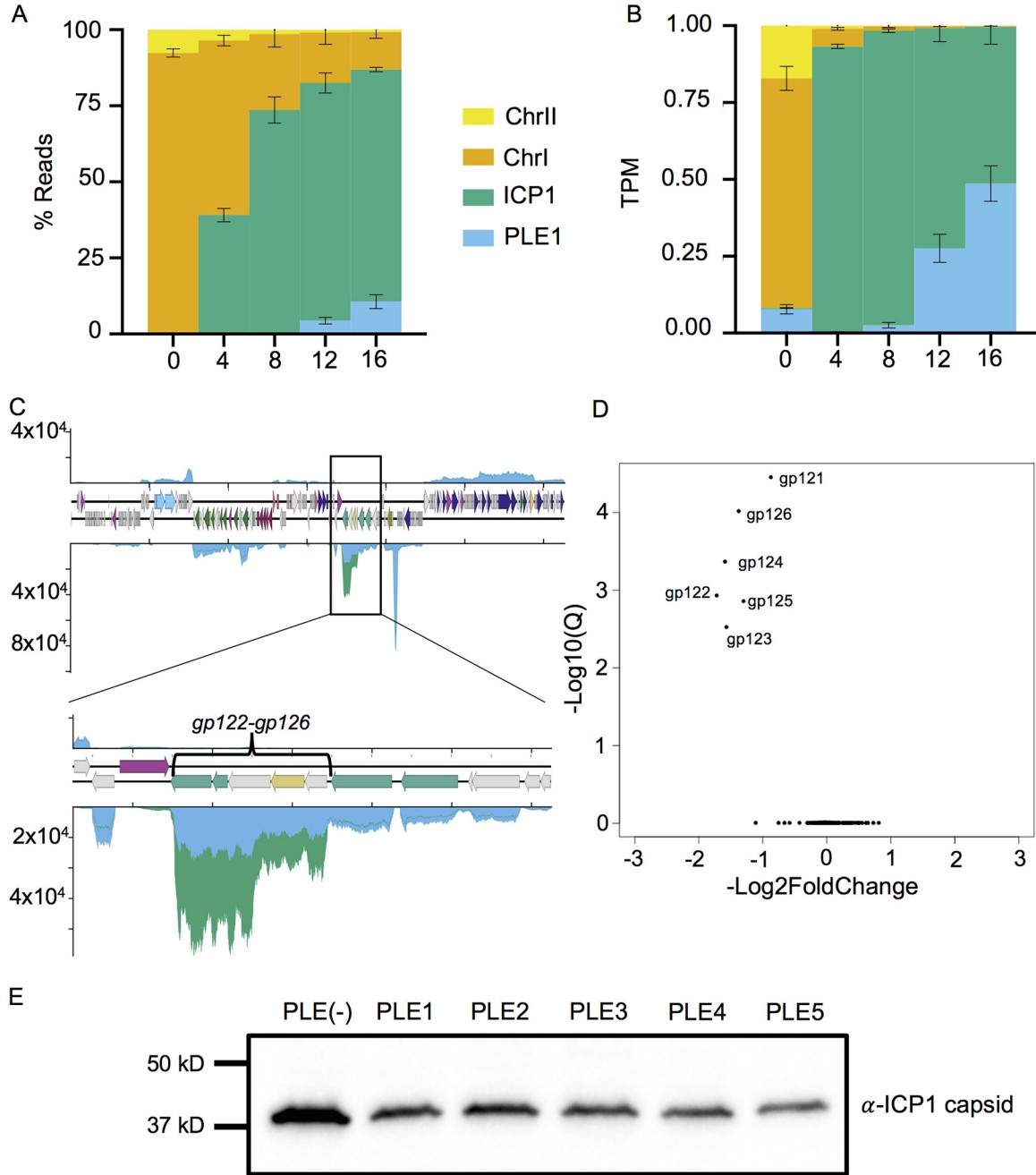

**FIG 8** PLE downregulates ICP1 capsid expression. (A) Percent read abundance for PLE1, *V. cholerae* chromosomes, and ICP1 over the infection time course. (B) Reads normalized to a TPM value based on the total number of reads from each element and the element's length. (C) Average relative coverage along ICP1's genome in PLE(−) (green) and PLE1 (blue) cultures at 16 min postinfection. The inset depicts ICP1's head morphogenesis operon. (D) Volcano plot of ICP1 differential gene expression in the PLE1 culture relative to that in the PLE(−) culture at 16 min postinfection. (E) Representative Western blot against Gp122, ICP1's major capsid protein from infections of PLE(−), PLE1, PLE2, PLE3, PLE4, and PLE5 cultures at 16 min postinfection. Quantification and replicates are shown in Fig. S7 in the supplemental material. For panels A to D, results incorporate values obtained from three biological replicates.

transcribed off the same strand of DNA that serves as the template for rolling circle replication. A block in ICP1 rolling circle replication would not impede ICP1 transcription if the newly synthesized DNA is not expressed but would permit PLE to interfere with ICP1 packaging, as has been hypothesized (24).

In addition to potential mechanistic explanations for how PLE is able to substantially restrict ICP1 replication without broad disruptions of ICP1 transcription, it is interesting to consider why such a discrepancy in PLE activity would be favorable. PLEs are not very

transcriptionally active until 8 min postinfection (Fig. 4A, Fig. S1, and Data Set S1). From 8 min postinfection onward, ICP1 transcripts largely comprise nucleotide metabolism genes, genes involved in DNA replication, and genes encoding virion structural components (Fig. 3). Aside from RepA, the replication initiation factor that directly interacts with the PLE origin of replication, PLE does not appear to encode dedicated replication machinery (24). Furthermore, PLE has been shown to rely on at least some ICP1 gene products for replication (24, 25). PLE also does not harbor identifiable structural genes and requires the same viral receptor as ICP1 for mobilization (15), suggesting that like other phage satellites (58), PLE is packaged within the same virion structural components as its host phage. The reliance on ICP1's virion production machinery incentivizes PLE to allow ICP1's transcriptional program to progress relatively unperturbed, since ICP1 is already producing the infrastructure for robust virion production. Our data suggest that rather than suppressing the production of ICP1 machinery and replacing it with PLEs' own, PLEs efficiently parasitize that machinery, redirecting it to PLEs' own genome and somehow excluding ICP1. Such a strategy would allow PLEs to benefit from ICP1's own reproductive adaptations while restricting ICP1 propagation among the host cell population.

While PLEs may benefit from permitting ICP1 gene expression, they also might benefit from retuning certain aspects of ICP1's transcriptional program to better fit their needs. Consistent with this hypothesis, we observed that one set of ICP1 genes has markedly decreased expression in the presence of all PLEs: *gp122* to *gp126* (Fig. 8C and D). This group of genes is predicted to be responsible for ICP1 capsid morphogenesis. The genes *gp122* and *gp123* are predicted to encode ICP1's major capsid protein and a capsid decoration protein, respectively, and *gp125* encodes a predicted protease, likely providing the proteolysis necessary for procapsid maturation as occurs in most tailed phages (59). Consistent with the predicted function of this cluster, these genes are robustly transcribed during late infection (Fig. 3), and Gp122 is detectable at 16 but not 8 min postinfection (see Fig. S6). Reduced expression of this gene cluster is seen in all PLEs, though the Q values we obtained from the differential analysis for PLE2 and PLE4 were less robust (see Fig. S7). Nevertheless, when the production of ICP1's major capsid protein was assessed in the presence of each PLE via Western blot, we found that, consistent with the differential transcription observed, all PLEs reduced ICP1 capsid production between 2- and 3-fold (Fig. 8E and Fig. S8). These results show that specific reduction in capsid production is a conserved activity among PLEs.

The PLEs' downregulation of ICP1 capsid genes may be evolutionarily tied to remodeling of the virion capsid. Capsid remodeling, more specifically, reduction in capsid dimensions, is a well-studied feature of phage satellite parasitism, having arisen in the phage satellite P4, the Gram-positive phage inducible chromosomal islands (PICIs), and the Gram-negative PICI's (58, 60), though it has yet to be shown for PLEs. Because capsid remodeling has been observed in the three other lineages of tailed phage satellites, capsid remodeling may be a general feature of phage satellite biology and is likely to be induced by PLEs. Capsid remodeling can restrict host viruses by assembling virions that are too small for the full-sized genome. Additionally, capsid remodeling likely increases the horizontal mobility of satellites. PLEs are not predicted to encode their own large terminase, suggesting that they rely on ICP1 packaging machinery in addition to structural components. Given that ICP1's genome is approximately 7 times the length of PLE, and ICP1 is predicted to package virions in a head-full fashion (24), packaging PLEs into ICP1-sized capsids would result in an overabundance of PLE genomes in a single transducing particle. Excess genome packaging would reduce the number of transducing particles that could be assembled for a set level of PLE genome replication.

While reducing capsid size would benefit PLE by reducing the amount of excess PLE DNA packaged per PLE transducing particle, further benefits could be gained by tuning gene expression to reduce production of excess capsid. Prior analyses have shown that protein translation is expected to be the most energy-intensive process for viruses within PLE and ICP1's genome size range (61, 62), and the major capsid transcript is by

far the most abundant mRNA produced by ICP1 (Fig. 3). Reducing the amount of capsid produced could help PLEs' recoup the costs of PLE gene expression and additional DNA replication, which do not occur during ICP1 infections in the PLE(−) background.

Much as capsid remodeling is a recognized occurrence among tailed phage satellites, our results reveal an emerging pattern of satellite elements tuning down the expression of phage late genes. Downregulation of capsid, at least relative to expression of other structural genes, has arisen independently among multiple phage satellites. Satellite P4 harbors a gene named *psu*, for polarity suppression unit. Psu acts as a coat decoration protein for the P2 capsid when remodeled by P4, increasing the stability of remodeled capsids (63). Additionally, Psu is well characterized as a Rho-binding antitermination factor (64–66). P2 regulates its structural genes through transcriptional attenuation: transcription frequently terminates before reading through the entire operon, and so the genes near the front of their operon, including those for the capsid and scaffold, are expressed to a greater extent than downstream genes (67). By preventing Rho-dependent termination, P4 retunes structural gene expression so that the ratio of capsid and scaffold relative to expression of other structural genes is reduced, with the ratios of capsid and scaffold to terminase being 5- to 10-fold lower in the presence of P4 (68). Initially, it was proposed that P4's tuning of structural gene transcription was the mechanism through which P2 capsids were remodeled; but later, remodeling was found to actually be caused by the Sid capsid scaffold, encoded by the same three-gene operon as Psu (69). Though Psu has undergone extensive biochemical characterization, little has been uncovered about the evolutionary importance of P4-induced antitermination since it was found to be dispensable for capsid remodeling.

Staphylococcal pathogenicity islands (SaPIs) provide an additional example of phage satellites repressing host phage genes. In the SaPI host phage 80α and the related phage 80, late genes are organized into a putative operon starting with the small terminase-encoding gene *terS*, followed by additional packaging genes, then head morphogenesis genes, tail morphogenesis genes, and finally, lysis genes. Some SaPIs have been found to repress *terS*, and it was inferred that there was also repression of the late gene operon (70). It was noted that repression must be incomplete, since SaPIs rely on phage structural, lysis, and large terminase genes for their propagation. While repression of *terS* could benefit SaPIs by preventing the packaging of host phage genomes, complete repression of the late operon would block SaPI particle production. Notably, only *terS* expression was measured for this operon. If internal promoters exist in this late operon, such that *terS* is silenced by SaPIs, head morphogenesis genes are repressed to an intermediate degree and tail and lysis genes are unaffected, it would be consistent with the pattern of head morphogenesis repression seen in P4 and PLE as well as the reproductive needs of SaPIs.

**Conclusions.** Here, we have provided the first study of phage satellite transcriptomics, obtaining transcriptional programs for ICP1 as well as all five variants of PLE. Aside from broadening our understanding of ICP1 and PLE biology, this work provides surprising insights into the biology of phage satellites. PLE integration alters the gene expression of other mobile elements in *V. cholerae*. Notably, one PLE increases the mobility of CTXΦ, showing that viral satellites can affect the spread of virulence genes that are harbored by unrelated mobile elements. More directly related to the PLE life cycle, we discovered that PLEs do not induce large-scale changes to ICP1's transcriptome, suggesting that PLE has adapted to take advantage of ICP1's lytic program as it occurs under conditions permissive to ICP1 replication. The notable exception is that PLEs downregulate ICP1's capsid morphogenesis operon, and this activity is conserved among PLEs and convergently evolved in the phage satellite P4. This pattern of evolution suggests a strong selective pressure for viral satellites to tune capsid expression, perhaps as a means to optimize resource use for satellite spread. It will be interesting to see if the patterns established here extend to other viral satellites and what other surprising aspects of viral satellite biology will emerge in the future.

## MATERIALS AND METHODS

**Strains and culture conditions.** All *V. cholerae* strains, including PLE(+) variants, used in this study are derived from E7946 to ensure comparisons in an otherwise isogenic background. Bacteria were grown on LB agar plates and in LB broth with aeration at 37°C. A detailed list of all strains used throughout this study can be found in Table S1, Tab 1, in the supplemental material. In continuity with previous PLE-related studies (15, 24), ICP1_2006_E engineered to lack CRISPR-Cas (ΔCRISPR Δcas2-3) (23) was used for all experiments. Phage titers were determined using a soft agar overlay method wherein ICP1 was allowed to adsorb to *V. cholerae* for 10 min at room temperature before the mixture was added to molten LB soft agar (0.3%) and poured onto 100-mm by 15-mm LB agar plates. Plaques were counted after overnight incubation at 37°C.

**Generation of mutant strains and constructs.** *V. cholerae* mutants were generated through natural transformation as described previously (71). For antibiotic-marked gene knockouts and overexpression constructs, splicing by overlap extension (SOE) PCR was used.

**Sample collection for RNA-seq.** Strains were grown to stationary phase in 2-ml cultures before being back diluted to an optical density at 600 nm ($OD_{600}$) of 0.05 in 6 ml LB broth. Strains were then grown to an $OD_{600}$ of 0.47 in 16- by 150-mm culture tubes with a Biochrom Ultrospec 10 (equivalent to an $OD_{600}$ of 0.3 with a 1-cm path length) before initial sample collection and phage infection at a multiplicity of infection (MOI) of 5. Immediately prior to infection, and then at 4, 8, 12, and 16 min postinfection, 1 ml of culture was taken and mixed with 1 ml of ice-cold methanol, before returning the remaining culture to the incubator. The sample and methanol mixtures were pelleted at $21,694 \times g$ at 4°C for 2 min, aspirated, washed with 1 ml ice-cold $1\times$ phosphate-buffered saline (PBS), and then pelleted and aspirated again. Pellets were snap-frozen in liquid $N_2$ and stored at −80°C until RNA isolation.

**RNA isolation.** RNA was extracted from samples using the Purelink RNA minikit (Thermo Fisher), and DNA was removed from isolated RNA samples using the TURBO DNA-Free kit (Thermo Fisher).

**cDNA library generation and sequencing.** RNA samples were submitted to the University of California Berkeley QB3 Core facility for cDNA library generation and sequencing. Ribosomal DNA was removed with an Illumina Ribo-Zero rRNA Removal kit (Bacteria) prior to cDNA generation. An S220 focused ultrasonicator (Covaris) was used to fragment the DNA, and library preparation was performed using the KAPA Hyper Prep kit for DNA (KK8504). Truncated universal stub adapters were used for ligation, and indexed primers were used during PCR amplification to complete the adapters and to enrich the libraries for adapter-ligated fragments. Samples were checked for quality on an AATI (now Agilent) fragment analyzer. Samples were then transferred to the Vincent J. Coates Genomics Sequencing Laboratory, another QB3-Berkeley Core Research Facility at UC Berkeley, where Illumina sequencing library molarity was measured with quantitative PCR with the Kapa Biosystems Illumina Quant qPCR Kits on a Bio-Rad CFX Connect thermal cycler. Libraries were then pooled evenly by molarity and sequenced on an Illumina HiSeq 4000 150 pared-end (PE) flow cell. Raw sequencing data were converted into fastq format in sample-specific files using the Illumina bcl2fastq2 software on the sequencing centers local Linux server system.

**RNA-seq analysis.** For each sample library, sequencing reads were mapped to separate *V. cholerae*, ICP1_2006E (ΔCRISPR Δcas2-3), and reference files as well as files for the appropriate PLE genomes in CLC Genomics Workbench version 12. Default RNA-seq mapping settings were used, with the exception that multiple mapping of individual reads was disabled. As noted previously (9, 11), RNA-seq of lytic infections possesses specific challenges because there are multiple genomes (two in most cases, three in the presence of a viral satellite such as PLE), undergoing changes in both their share of total transcripts in culture and the relative expression of their genes. To address this, gene expression was normalized on a per genome basis for differential expression analysis. Differential expression analysis was performed using the DESeq2 (29) R/Bioconductor package with default parameters. For data visualization, heat maps of $log_2$ TPM values were plotted using the aheatmap function from the NMF R package. Volcano plots were generated using the EnhancedVolcano package and function. Our read counts and DESeq2 results are provided in Data Set S1. The accession numbers for reference sequences used for mapping can be found in Table S1 (Tab 5).

For the generation of reads tracks, RNA-seq reads were mapped to the reference sequences using bowtie2 v2.3.4.1 (72), with the following settings: "– end-to-end –very-sensitive –no-unal –no-mixed –no-discordant." For each sample, read coverage was normalized to sequencing depth, and replicates were then averaged.

**CTX transduction assays.** *V. cholerae* PLE(−), PLE1, and PLE2 CTX(+) donor strains were modified by replacing *ctxAB* with a kanamycin resistance cassette. CTXΦ production was induced in these strains by growing up to an $OD_{600}$ of 0.3 followed by a 16-h incubation at 37°C with aeration in LB supplemented with mitomycin C (20 ng/ml) (Sigma). *V. cholerae* CTX(−) recipient strains were engineered to harbor a cassette inserted in the *lacZ* locus containing a spectinomycin resistance gene and *toxT* under the control of $P_{tac}$ and a theophylline-inducible riboswitch. Recipient strains were grown to an $OD_{600}$ of 0.3 and then induced with addition of 1.5 mM theophylline and 1 mM isopropyl-β-D-thiogalactopyranoside (IPTG) at 37°C for 16 h in LB plus 10 mM $MgCl_2$ without agitation. After mitomycin C treatment, donor strains were centrifuged for 3 min at $5,000 \times g$ twice to ensure maximum removal of donor cells from CTXΦ-containing supernatant. Cleared donor supernatants were mixed with recipient cultures 1:4 and incubated at 37°C for 1 h without agitation. Transduction mixtures were plated on LB plates supplemented with kanamycin (75 μg/ml) and spectinomycin (100 μg/ml) and incubated overnight at 37°C to quantify transductants.

**Western blots.** Isogenic *V. cholerae* strains either lacking PLE or with an integrated PLE (PLE1 to -5) were grown to an $OD_{600}$ of 0.3 and infected with ICP1_2006E ΔCRISPR Δ*cas2-3* at an MOI of 1 and returned to the incubator at 37°C with aeration. At 16 min after phage addition, 1 ml of infected culture was collected and mixed with an equal volume of ice-cold methanol. Samples were centrifuged at $5,000 \times g$ for 10 min at 4°C to pellet infected cells. Pellets were washed once with ice-cold PBS and resuspended in lysis buffer (50 mM Tris, 150 mM NaCl, 1 mM EDTA, 0.5% Triton X-100, 1× protease inhibitor [Thermo Pierce Protease and Phosphatase inhibitor tablet]). Protein concentration was quantified with a Pierce bicinchoninic acid (BCA) protein assay kit (Thermo). Thirty micrograms of total protein sample was mixed with Laemmli buffer (Bio-Rad) and boiled at 99°C for 10 min. Samples were run on Any-kD TGX-SDS-PAGE gels (Bio-Rad) and transferred to nitrocellulose membranes with a Transblot Turbo Transfer system (Bio-Rad). Custom primary peptide antibody generated in rabbits against ICP1 capsid (Gp122, YP_004251064.1) (GenScript) was diluted 1:1,500. Band detection was conducted with a goat anti-rabbit IgG horseradish peroxidase (HRP) secondary antibody (Bio-Rad) at 1:10,000 followed by development with Clarity Western ECL substrate (Bio-Rad) and imaging on a ChemiDoc XRS imaging system (Bio-Rad).

**Data availability.** Sequence data for samples used in this work can be found in the Sequence Read Archive under the BioProject accession PRJNA609114.

## SUPPLEMENTAL MATERIAL

Supplemental material is available online only.

**FIG S1**, PDF file, 1.9 MB.
**FIG S2**, PDF file, 0.7 MB.
**FIG S3**, PDF file, 1.4 MB.
**FIG S4**, PDF file, 0.3 MB.
**FIG S5**, PDF file, 0.2 MB.
**FIG S6**, PDF file, 0.5 MB.
**FIG S7**, PDF file, 0.3 MB.
**FIG S8**, PDF file, 0.5 MB.
**TABLE S1**, XLSX file, 0.1 MB.
**DATA SET S1**, XLSX file, 3.9 MB.

## ACKNOWLEDGMENTS

We thank Andrew Camilli for sharing the CTX minus recipient strain. We thank the University of California Berkeley QB3 Core facility for cDNA library preparation and sequencing. We thank all members of the Seed lab past and present for useful discussions and, particularly, Kristen LeGault and Caroline Boyd for critical feedback. We also thank Bob Bender, Matt Chapman, and Lyle Simmons for helpful discussions.

This project was supported by the National Institute of Allergy and Infectious Diseases (R01AI127652 to K.D.S.); K.D.S. is a Chan Zuckerberg Biohub Investigator and holds an Investigators in the Pathogenesis of Infectious Disease Award from the Burroughs Wellcome Fund. This research was also funded in part by a National Institutes of Health NRSA Trainee appointment on grant number 5 T32 GM 132022 (to Z.N.).

The funders had no role in study design, data collection and interpretation, or the decision to submit the work for publication.

Z.K.B., Z.N., and K.D.S. conceptualized the study and designed experiments. Z.K.B. and P.B. performed phage infections and sample collection for RNA sequencing. Z.K.B. generated strains for this study and performed differential expression analyses. Z.N. performed Western blotting and transduction experiments. A.A. contributed to data analysis and visualization of RNA-seq data. Z.K.B. wrote the manuscript with revisions provided by K.D.S. All authors discussed the results and commented on and approved the final manuscript.

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
