## [Reviewer comments · mSystems]

A family of viral satellites manipulates invading virus gene expression and can affect cholera toxin mobilization

Zachary Barth, Zoe Netter, Angus Angermeyer, Pooja bhardwaj, and Kimberley Seed

Corresponding Author(s): Kimberley Seed, University of California, Berkeley

Review Timeline:

Submission Date:	April 22, 2020
Editorial Decision:	June 16, 2020
Revision Received:	September 23, 2020
Accepted:	September 24, 2020

Editor: Seth Bordenstein

Reviewer(s): The reviewers have opted to remain anonymous.

Transaction Report:

DOI: <https://doi.org/10.1128/mSystems.00358-20>

June 16, 2020

Dr. Kimberley D Seed
University of California, Berkeley
Department of Plant & Microbial Biology
271 Koshland Hall
Berkeley, CA 94720

Re: mSystems00358-20 (A family of viral satellites manipulates invading virus gene expression and affects cholera toxin mobilization)

Dear Dr. Kimberley D Seed:

Below you will find the comments of two experts in the area of the manuscript submitted to mSystems. Both agreed the paper is on track for publication, and my recommendation based on their thorough feedback is to accept the article pending relatively moderate modifications. Each reviewer carefully read the article and provided detailed comments and ideas for alternative or conservative interpretations. I also recommend that you find ways to consolidate figures and move several of the 17 supplementary figures to the main text as the journal prefers to have data in the main text when possible in order to make the article more readable and transparent. Ideally, the number of supplementary figures should be reduced by half or more. Congratulations, and we look forward to reading the revision.

To submit your modified manuscript, log onto the eJP submission site at <https://msystems.msubmit.net/cgi-bin/main.plex>. If you cannot remember your password, click the "Can't remember your password?" link and follow the instructions on the screen. Go to Author Tasks and click the appropriate manuscript title to begin the resubmission process. The information that you entered when you first submitted the paper will be displayed. Please update the information as necessary. Provide (1) point-by-point responses to the issues raised by the reviewers as file type "Response to Reviewers," not in your cover letter, and (2) a PDF file that indicates the changes from the original submission (by highlighting or underlining the changes) as file type "Marked Up Manuscript - For Review Only."

Due to the SARS-CoV-2 pandemic, our typical 60 day deadline for revisions will not be applied. I hope that you will be able to submit a revised manuscript soon, but want to reassure you that the journal will be flexible in terms of timing, particularly if experimental revisions are needed. When you are ready to resubmit, please know that our staff and Editors are working remotely and handling submissions without delay. If you do not wish to modify the manuscript and prefer to submit it to another journal, please notify me of your decision immediately so that the manuscript may be formally withdrawn from consideration by mSystems.

To avoid unnecessary delay in publication should your modified manuscript be accepted, it is important that all elements you upload meet the technical requirements for production. I strongly recommend that you check your digital images using the Rapid Inspector tool at <http://rapidinspector.cadmus.com/RapidInspector/zmw/>.

If your manuscript is accepted for publication, you will be contacted separately about payment

when the proofs are issued; please follow the instructions in that e-mail. Arrangements for payment must be made before your article is published. For a complete list of **Publication Fees**, including supplemental material costs, please visit our website.

Sincerely,

Seth Bordenstein

Editor, mSystems

Journals Department
Reviewer comments:

Reviewer #1 (Comments for the Author):

The work presented in "A family of viral satellites manipulates invading virus gene expression and affects cholera toxin mobilisation" studies the transcriptomic profile of a strain of *Vibrio cholerae* in different conditions related to mobile genetic elements. It brings new information on the interactions between different mobile elements, some of which could be important for the pathogenic conversion of environmental *V. cholerae* strains.

The manuscript is well organized and very well written. However, some paragraphs in the Results section are very speculative. I would suggest adding a Discussion section. In addition, the title suggests that PLEs have a role in CTX production, whereas the effect is limited to PLE1. Indeed, the other PLEs seem to increase the *rstR* expression, which is expected to block their replication. Therefore, I suggest that the authors change their title.

Regarding CTX mobilisation, I wonder if the infection by ICP1 might not induce the SOS response. What happens with CTX in this case?

Specific points:

1) Regarding CTX production, I wonder if the infection by ICP1 might not induce the SOS response. What happens with CTX in this case?

2) In the conditions used, PLEs do not induce large scale changes to ICP's transcriptome. However, I wonder if this could not be due to the MOI for the experiments (MOI=5). Is this MOI close to what normally occurs in *Vibrio cholerae*'s natural environment? In other works, the MOI used is only 0.1.

3) Could the authors inform the accession number of the strain used? I downloaded CP047303 (strain E7946) and the position of the different ORFs do not correspond to the positions informed in the supplementary files

For example:

In NoPLE_reads_count, tyrA -> 743364..744491

In CP047303, tyrA -> 627864..628991

4) Have the authors tried to analyze their data with edgeR, as in Howard Varone et al, 2019. Phage-specific metabolic reprogramming of virocells. Could Poisson mixed-effect model be used to quantify transcript-level gene expression in RNA-Seq?

5) "Line 266: PLEs show some expression of int

I would like to know if PLEs are stably integrated or if they could be also found as extrachromosomal.

6) "Line 293-294: Read counts for PLE genes remain low at 4 minutes post infection (S1 Data), and then at 8 minutes post infection transcriptional patterns start to emerge

At 8 minutes, 80% of the reads are from ICP1. Taking into account that there are also reads from the cells, what is the proportion of reads from PLE? What happens with a lower MOI?

7) "Line 329-341: Non-coding RNAs flanked by terminal inverted repeats often occur in miniature inverted-repeat transposable elements (MITEs), and these have been examined in several bacterial species (48). This similarity suggests that the PLE IIR may have evolved from a selfish mobile ancestor. The PLE IIR transcript also has antisense homology to the leader sequences of several PLE ORFs (S10B Fig). Complementarity between a ncRNA and gene leader sequences is seen in the regulatory RNAs from phages P1, P7, and N15, as well as the phage satellites P4 and fR73 (49-51). With the exception of N15's RNA which promotes the lytic cycle, these regulatory RNAs function to repress the lytic cycle. These roles appear unlikely in PLE, since there is an accumulation of the PLE IIR transcript as infection progresses, but it is not abundant until 12 minutes post infection, which is after PLE early gene expression (Fig 3B, S2-S5 Figs). The PLE IIR transcript's expression pattern appears more consistent with modifying gene expression during infection rather than acting as a global activator or repressor of PLE activity."

The paragraph is quite complex: It starts by proposing that the PLE IIR could be a selfish mobile element, and then rambles about N15, stating that PLE IIR are probably not involved in the lytic cycle. However, the paragraph ends on the observation that PLE IIR transcript's expression pattern modifies gene expression during infection. This is rather confusing and should be written differently

8) "Lines 361-364 In *V. cholerae*, both mobile genetic elements and genes in the superintegron are known to be repressed by the bacterial chromatin protein H-NS (53-55). An alternative explanation for why we see changes in superintegron gene expression is that PLE integration alters the nucleoid architecture of this region through its own recruitment of H-NS"

To my knowledge, it is known that H-NS affects DNA conformation locally, however it has not been demonstrated that the architecture modifies gene expression. Can the authors add a citation to support the alternative explanation?

An assay showing the changes in H-NS binding pattern (Chip-Seq) could help to shed light on the process.

9) "Lines 365-367 Notably, many of the superintegron genes are multi-copy, so the number of differentially regulated genes may be over-reported if the reads mapping cannot differentiate reads from multi-copy genes from different loci"

I would like to have more detail about the mapping and the analysis. How can they differentiate

RNA from duplicated genes? If a gene is duplicated and transcription increases 2 times, is it possible to say that at the end there wasn't a change?

Later, they indicate in a volcano graph that "tlcR1" and "tlcR2". Are the genes TlcR1 and tlcR2 different enough to be differentiated ?

10) S11 FIG In PLE4 and PLE5 graph's there are other blue points corresponding to IMEX's genes over expression. What are those?

11) "Line 390. These results reveal potentially far reaching effects that mobile elements can have on each other, as well as the hosts they share"

It seems very clear that there are interactions between mobile elements and their host. But regarding the effect on CTX mobility, is there any report associating PLE1 with pathogenicity? Is VC containing PLE1 more pathogenic than VC containing PLE2?

12) "Line 424 PLE also does not encode identifiable structural genes, and requires the same viral receptor as ICP1 for mobilization (14), suggesting that like other phage satellites (57), PLE is packaged within the same virion structural components as its host -"

It is not known that PLE are packaged using ICP1 machinery neither if it encodes a protein to reshape the virion, like other phage inducible islands. Even if there are a variety of capsid re-direction mechanisms (Dokland 2019, Molecular Piracy: Redirection of Bacteriophage)

13) "Capsid Assembly by Mobile Genetic Elements"

did the authors tried to do a bioinformatic analysis to find possible factors on PLE involved on those functions?

14) "Line 431: PLEs efficiently parasitize that machinery, redirecting it to PLE's own genome and somehow excluding ICP1"

Even though it makes sense that PLE do not shut down completely ICP1 transcription, as I mentioned before, could this observation be due to the MOI used?

15) Legend of Figure 7: (C) and (D) captions are inverted.

16) "Line 471: Reducing the amount of capsid produced could help PLEs' recoup the costs of PLE gene expression and additional DNA replication which do not occur during ICP1 infections in the PLE(-) background"

I don't know if it is the case for ICP1, but some phages encode factors that modify the metabolism of the host. Is the extra cost of replicating PLE be significant compared to ICP1 viral particles production?

Reviewer #2 (Comments for the Author):

Barth et al. have written a compelling article on the transcriptome of *V. cholerae* infected with ICP1 (type strain and associated variants). The authors examined the hypothesis that Phage Inducible Chromosomal Island-Like Elements (PLEs) inhibit the expression of ICP1 genes. Among several findings, they found inhibition of a major capsid morphogenesis operon, and PLEs were associated with increased cholera toxin phage gene expression (replication, integration and mobilization) with SOS induction.

Major comments

The manuscript and analysis were well performed experimentally with respect to the rigor of collection, processing and sequencing. This is difficult given the short time intervals in the lytic cycle.

1. I appreciated the Western Blot of Gp122. Two important items that are missing are: (i) what did the Western Blots for Gp122 show at the earlier time points (PLE + and PLE-). These data should be negative and will validate the sequencing data and confirm the late gene findings. (ii) The Supplemental Figure S17C should present statistical analysis of the data shown.
2. Despite the Gp122 work and ctx phage experiment, experimental validations of the bio-informatic findings need molecular proofs to move the findings forward. For example, can the authors postulate why motility was upregulated? Were there behavioral phenotypes observed that might support this transcriptional finding? Are there qPCR assays that can be done to confirm the RNA seq on a few of these items found? These items can be addressed in part herein and/or pushed to a follow up study.

Minor comments:

1. Is ICP1_2006 considered the 'type' strain? What was the clinical context / scientific rationale for picking this strain? Line 505
2. Was there an attempt to repeat the collection at a lower O.D. before quorum sensing triggers in order to differentiate density dependent transcriptional events? Consider adding to the discussion.
3. Was induction of ctx phage observed in AKI media (1.5% Bacto-Peptone [Difco], 0.4% yeast extract, 0.5% NaCl). If no, why not. Line 507.
4. Add company name/ core lab that made rabbit anti ICP1 capsid. Line 597.
5. The estimate that that 25% of reads at 4 minutes are ICP1 transcripts is impressive (line 166). With respect to VC, why is there a lack of balance between the reads from the two VC chromosomes?

We thank the editor and reviewers for their insightful critiques of our manuscript. We have addressed their comments and have incorporated the suggestions as outlined in this response.

In response to the editor's suggestion to reduce our number of supplemental figures, we have moved supplemental figure 1 into the main text, reorganized the data in some of our supplemental figures, and consolidated several of our supplemental figures by renumbering them. The large number of supplemental figures came about from us wanting to show that the expression patterns observed for PLE1 and presented in figures 2, 4, 6 and 8 of the main text are representative for the four other PLEs (presented in the supplement). We believe these figures should remain supplementary, since they are presenting similar data to what is in the main text in different strain backgrounds. They serve as evidence that the trends presented in the paper are generalizable across the different PLEs, and we do not think they are necessary for the average reader to understand our manuscript. At the same time, we want to provide this data in a legible form for readers who would like to see them. We think the best solution is to present the data as supplementary figures (the S1, S2, and S7 figures) that span a few pages, rather than consolidating the panels onto a single page (which would make them too small to see in our opinion). Setting up these figures this way allows the inclusion of interpretable high resolution images, while reflecting the relative importance of these figures to supporting our main points. We hope this is a satisfactory solution.

For reviewer comments, we have addressed their points individually below (comments in bold, responses in regular font):

Reviewer #1 (Comments for the Author):

The work presented in "A family of viral satellites manipulates invading virus gene expression and affects cholera toxin mobilisation" studies the transcriptomic profile of a strain of *Vibrio cholerae* in different conditions related to mobile genetic elements. It brings new information on the interactions between different mobile elements, some of which could be important for the pathogenic conversion of environmental *V. cholerae* strains.

The manuscript is well organized and very well written. However, some paragraphs in the Results section are very speculative. I would suggest adding a Discussion section. In addition, the title suggests that PLEs have a role in CTX production, whereas the effect is limited to PLE1. Indeed, the other PLEs seem to increase the *rstR* expression, which is expected to block their replication.

Therefore, I suggest that the authors change their title.

Regarding CTX mobilisation, I wonder if the infection by ICP1 might not induce the SOS response. What happens with CTX in this case?

Rather than separate Results and Discussion sections, the manuscript has a combined Results and Discussion section (line 147), as we have seen done for several papers published in *mSystems*. We found that this was the best way to present the experimental results and their possible implications in the most succinct manner

possible. If we included a separate discussion section, we would have to reiterate our results, and thus run into considerable space constraints. We have demarcated our speculative statements with qualifying language, and are open to further language adjustments if deemed necessary.

We agree with the reviewer comment that the CTX transduction phenotype is limited to PLE 1 and we have therefore modified the title to 'A family of viral satellites manipulates invading virus gene expression and can affect cholera toxin mobilization', adding the word 'can' to specify that boosting cholera toxin mobilization is not a conserved feature of PLEs.

We do not see evidence of ICP1 inducing the SOS response in the transcriptome data. To address this point, we have added the following sentences: The SOS response to DNA damage is often induced by phage infection, and several genes under SOS regulation have been identified in *V. cholerae*(37). We did not observe differential expression of genes under the SOS regulon following infection, suggesting that ICP1 may have a mechanism to avoid or repress this response. (Lines 221-224)

We also did not observe any notable effect on CTX Φ gene expression during ICP1 infection, which is now explicitly stated (Line 222). While the very low CTX Φ expression in uninfected cells coupled with the progressive depletion of *V. cholerae* gene expression makes us reluctant to make definitive conclusions regarding ICP1 infection and CTX Φ induction, there is no transcriptomic data that warrants further experimental investigation of CTX mobilization during ICP1 infection in this manuscript.

Specific points:

1) Regarding CTX production, I wonder if the infection by ICP1 might not induce the SOS response. What happens with CTX in this case?

Please see our previous points, we do not see any indication of this induction from the analyses performed.

2) In the conditions used, PLEs do not induce large scale changes to ICP's transcriptome. However, I wonder if this could not be due the MOI for the experiments (MOI=5). Is this MOI close to what normally occurs in Vibrio cholera's natural environment? In other works, the MOI used is only 0.1.

While we agree that looking at transcription in low MOI infection conditions could be an interesting line of inquiry, we designed our experiments to best reveal the details of ICP1's transcriptional program in an infected cell during one round of replication. Since RNA-seq reads are averaged across the culture population, the best way to have our sequence data reflect the profile of an infected cell is to ensure that the majority of cells in culture are infected. A low MOI infection would have a high background of uninfected host reads during the initial infection, obscuring the infection transcriptome, and precluding interpretation of PLE's and *V. cholerae*'s response to infection. In most of the transcriptomic studies we have cited, (refs 6,7,9,11,12, and 13) cultures for RNA-seq

are infected at an MOI of 5 or higher, so our methods are well aligned with previous studies.

3) Could the authors inform the accession number of the strain used? I downloaded CP047303 (strain E7946) and the position of the different ORFs do not correspond to the positions informed in the supplementary files

For example:

In NoPLE_reads_count, tyrA -> 743364..744491

In CP047303, tyrA -> 627864..628991

We include these accession numbers in table S5. They are NZ_CP024162 and NZ_CP024163 for E7946. The location of the accession numbers is noted in our Materials and Methods section (Lines 640-642).

4) Have the authors tried to analyze their data with edgeR, as in Howard Varone et al, 2019. Phage-specific metabolic reprogramming of virocells. Could Poisson mixed-effect model be used to quantify transcript-level gene expression in RNA-Seq?

We did perform preliminary analysis with edgeR, and similar results were obtained to what we present in the paper. Established convention for bacteriophage transcriptomic analysis papers appears to be presenting DGE analysis done with a single tool. EdgeR and DESeq2 appear to be the most popular tools, and DESeq2 appears to be more common in phage transcriptomic papers (several examples are cited in line 76). We have added the Howard-Varona paper to the examples we cite.

It is not immediately clear to us what the benefits of using a Poisson mixed-effect model would be for analysis of our system. We do discuss the limitations of using traditional DGE tools to analyze lytic infection (lines 160-166). While it would be an exciting project to develop models for better analysis of transcriptional changes during phage infection, our intention with this manuscript is to glean new insights by applying established analysis techniques to our specific system. We think we have been successful in that regard. Other groups are welcome to analyze our data set further if they wish to develop other analysis pipelines for phage infection transcriptomic datasets, the raw data is available in the SRA (Line 648).

**5) "Line 266: PLEs show some expression of int
I would like to know if PLEs are stably integrated or if they could be also found as extrachromosomal.**

We've seen no evidence of excision in PLE bearing strains in the absence of phage infection. A study of PLE integration and excision can be found in ref 23, McKitterick and Seed 2018. In short, PLE 1, 3, 4 & 5 excision requires an ICP1-encoded protein, so those PLEs are stably integrated and excision is not observed in uninfected cells (this is described in lines 123-125). Evidence suggests PLE 2 also requires an ICP1 encoded protein for excision, however the identity of this protein is not currently known. Our

finding that Int is expressed in uninfected cells (Line 289) is consistent with previous investigation of Int which is detectable by Western Blot in uninfected cells (Ref 23). We now state in the manuscript "Expression of *int* in uninfected cells is consistent with previous work where Int was detectable by Western blot in the absence of ICP1 (23)." (Line 291)

6) "Line 293-294: Read counts for PLE genes remain low at 4 minutes post infection (S1 Data), and then at 8 minutes post infection transcriptional patterns start to emerge

At 8 minutes, 80% of the reads are from ICP1. Taking into account that there are also reads from the cells, what is the proportion of reads from PLE? What happens with a lower MOI?

We have not performed a low MOI infection because reads from uninfected cells would predominate in our culture (as stated in our previous reply). The proportion of PLE reads at these time points can be seen in Figure 7. PLE is very lowly expressed in uninfected *V. cholerae* cells, as is depicted in Figure 7 A and B.

7) "Line 329-341: Non-coding RNAs flanked by terminal inverted repeats often occur in miniature inverted-repeat transposable elements (MITEs), and these have been examined in several bacterial species (48). This similarity suggests that the PLE IIR may have evolved from a selfish mobile ancestor. The PLE IIR transcript also has antisense homology to the leader sequences of several PLE ORFs (S10B Fig). Complementarity between a ncRNA and gene leader sequences is seen in the regulatory RNAs from phages P1, P7, and N15, as well as the phage satellites P4 and fR73 (49-51). With the exception of N15's RNA which promotes the lytic cycle, these regulatory RNAs function to repress the lytic cycle. These roles appear unlikely in PLE, since there is an accumulation of the PLE IIR transcript as infection progresses, but it is not abundant until 12 minutes post infection, which is after PLE early gene expression (Fig 3B, S2-S5 Figs). The PLE IIR transcript's expression pattern appears more consistent with modifying gene expression during infection rather than acting as a global activator or repressor of PLE activity."

The paragraph is quite complex: It starts by proposing that the PLE IIR could be a selfish mobile element, and then rambles about N15, stating that PLE IIR are probably not involved in the lytic cycle. However, the paragraph ends on the observation that PLE IIR transcript's expression pattern modifies gene expression during infection. This is rather confusing and should be written differently

This section has been rewritten to remove extraneous details and make the main ideas clearer. It now reads as 'In each PLE, the most abundant transcript is located between C_{R1} and C_{R2} (Fig 6 and S2 Fig). This transcript occurs between a set of inverted repeats, and for this reason, we have tentatively named the transcript the inter-inverted repeat (IIR) transcript (S3 Fig). The PLE IIR transcript also has antisense homology to the leader sequences of several PLE ORFs (S3B Fig). Complementarity between a

ncRNA and gene leader sequences is seen in the regulatory RNAs from phages P1, P7, and N15, as well as the phage satellites P4 and ϕ R73 (52–54), suggesting that the IIR transcript may have a role in regulating PLE gene expression. Non-coding RNAs flanked by terminal inverted repeats often occur in miniature inverted-repeat transposable elements (MITEs) (55), suggesting that the PLE IIR transcript and the aforementioned phage RNAs may have been ‘domesticated’ from a class of mobile genetic elements.”

8) "Lines 361-364 In *V. cholerae*, both mobile genetic elements and genes in the superintegron are known to be repressed by the bacterial chromatin protein H-NS (53-55). An alternative explanation for why we see changes in superintegron gene expression is that PLE integration alters the nucleoid architecture of this region through its own recruitment of H-NS"

To my knowledge, it is known that H-NS affects DNA conformation locally, however it has not been demonstrated that the architecture modifies gene expression. Can the authors add a citation to support the alternative explanation? An assay showing the changes in H-NS binding pattern (Chip-Seq) could help to shed light on the process.

While H-NS is known to remodel local DNA architecture and the H-NS regulon includes genes that aren't directly occluded by H-NS, we were unable to find a citation that experimentally shows gene regulation by H-NS through conformational changes of proximal DNA. While others have suggested this possibility, and we think it could explain the PLEs' observed effects on gene expression, we recognize the reviewer's concern that this explanation may be overly speculative, so we have removed those lines from our manuscript.

We agree that Chip-Seq would be a good way to address our hypothesis regarding H-NS, but that is clearly beyond the scope of the current manuscript.

9) "Lines 365-367 Notably, many of the superintegron genes are multi-copy, so the number of differentially regulated genes may be over-reported if the reads mapping cannot differentiate reads from multi-copy genes from different loci" I would like to have more detail about the mapping and the analysis. How can they differentiate RNA from duplicated genes? If a gene is duplicated and transcription increases 2 times, is it possible to say that at the end there wasn't a change?

Later, they indicate in a volcano graph that "tlcR1" and "tlcR2". Are the genes TlcR1 and tlcR2 different enough to be differentiated ?

This is partially addressed in our methods section where we write “Default RNA-seq mapping settings were used, with the exception that multiple mapping of individual reads was disabled.” With multiple mapping disabled, the observed changes in gene expression should be real, though it may be impossible to say which DNA copy the increased transcripts are coming from. The duplication is also present in both conditions

that we are comparing, so without multiple mapping, there shouldn't be inflation of the genes' transcripts.

The two *tlcR* genes are identical, but there is a transposon inserted ~300bp downstream of *tlcR2*. With 150x150 sequencing and an average insert size of 450bp, this should allow the 3' end of the genes to be differentiated, while reads mapping to the 5' end will likely be assigned randomly between the two genes.

10) S11 FIG In PLE4 and PLE5 graph's there are other blue points corresponding to IMEX's genes over expression. What are those?

These are TLC genes of unknown function with predicted DNA binding domains. This information is provided in S4 data.

11) "Line 390. These results reveal potentially far reaching effects that mobile elements can have on each other, as well as the hosts they share"

It seems very clear that there are interactions between mobile elements and their host. But regarding the effect on CTX mobility, is there any report associating PLE1 with pathogenicity? Is VC containing PLE1 more pathogenic than VC containing PLE2?

There are no reports of associating PLE1 with increased pathogenicity, nor are there published experiments evaluating the pathogenicity of PLE1 vs PLE2 hosts in animal models. Thus far, PLEs have only been found in pandemic isolates of *V. cholerae* (those that encode CTX and the O1 antigen as opposed to the many times greater number of avirulent environmental lineages), but of course many pandemic *V. cholerae* strains do not encode PLEs (ie refs 15 and 28), indicating they are not required for virulence. We do not have any evidence that PLEs directly confer a virulence phenotype, and based on gene content, we don't expect them to. Our results here indicate that CTX mobility is enhanced in PLE1+ strains, though we did not see upregulation of cholera toxin for example in the presence of PLE1 under the conditions evaluated.

12) "Line 424 PLE also does not encode identifiable structural genes, and requires the same viral receptor as ICP1 for mobilization (14), suggesting that like other phage satellites (57), PLE is packaged within the same virion structural components as its host-"

It is not known that PLE are packaged using ICP1 machinery neither if it encodes a protein to reshape the virion, like other phage inducible islands. Even if there are a variety of capsid re-direction mechanisms (Dokland 2019, Molecular Piracy: Redirection of Bacteriophage)

13) "Capsid Assembly by Mobile Genetic Elements"

did the authors tried to do a bioinformatic analysis to find possible factors on PLE involved on those functions?

We are under the impression that 12 and 13 were meant to be one point. We apologize if this is a mistaken assumption.

Unfortunately, capsid scaffolds do not present reliable bioinformatics signatures, so we do not have a directed way to look for this. However, our lab is currently experimentally investigating this aspect of the PLE lifecycle, stay tuned.

**14) "Line 431: PLEs efficiently parasitize that machinery, redirecting it to PLE's own genome and somehow excluding ICP1"
Even though it makes sense that PLE do not shut down completely ICP1 transcription, as I mentioned before, could this observation be due to the MOI used?**

To our knowledge, there is no evidence that viral transcription within an infected cell is impacted by MOI. Since PLE does not appear to activate until ICP1 replication has begun, we are skeptical that PLE would be more repressive towards ICP1 if the starting MOI were lower, though we cannot exclude the possibility that MOI could influence the outcome of the infection. However, as the reviewer has pointed out, we do not address varying the MOI or other conditions (like media, temperature ect) in this paper, and that remains an area of potential future investigation.

15) Legend of Figure 7: (C) and (D) captions are inverted.

We thank the reviewer for pointing out this error. It has been fixed.

**16) "Line 471: Reducing the amount of capsid produced could help PLEs' recoup the costs of PLE gene expression and additional DNA replication which do not occur during ICP1 infections in the PLE(-) background"
I don't know if it is the case for ICP1, but some phages encode factors that modify the metabolism of the host. Is the extra cost of replicating PLE be significant compared to ICP1 viral particles production?**

Our transcriptomic results suggest that widespread metabolic changes are occurring in the cell during ICP1 infection.

Back of the envelope calculations suggest that more new DNA is being synthesized in a PLE(+) infection than a PLE(-) infection. We have not performed a formal analysis to look at this, and with the tools we have available doing so would be difficult.

Reviewer #2 (Comments for the Author):

Barth et al. have written a compelling article on the transcriptome of *V. cholerae* infected with ICP1 (type strain and associated variants). The authors examined the hypothesis that Phage Inducible Chromosomal Island-Like Elements (PLEs) inhibit the expression of ICP1 genes. Among several findings, they found inhibition of a major capsid morphogenesis operon, and PLEs were associated with increased cholera toxin phage gene expression (replication, integration and mobilization) with SOS induction.

Major comments

The manuscript and analysis were well performed experimentally with respect to the rigor of collection, processing and sequencing. This is difficult given the short time intervals in the lytic cycle.

1. I appreciated the Western Blot of Gp122. Two important items that are missing are: (i) what did the Western Blots for Gp122 show at the earlier time points (PLE + and PLE-). These data should be negative and will validate the sequencing data and confirm the late gene findings. (ii) The Supplemental Figure S17C should present statistical analysis of the data shown.

We have included an Gp122 western blot 8 minute time point in the new figure S6, as expected there is no signal for Gp122 at this early time point. We have added statistical analysis for quantification of the Western blots as requested (Line 1164)

2. Despite the Gp122 work and ctx phage experiment, experimental validations of the bio-informatic findings need molecular proofs to move the findings forward. For example, can the authors postulate why motility was upregulated? Were there behavioral phenotypes observed that might support this transcriptional finding? Are there qPCR assays that can be done to confirm the RNA seq on a few of these items found? These items can be addressed in part herein and/or pushed to a follow up study.

We agree that these findings will ultimately have to be validated and studied further. Because we do not know of reference genes that are stably expressed through ICP1 infection, we don't believe that we would be able to accurately measure transcript level differences via qPCR. This seems to be a problem for any lytic phage infected culture, and we are unaware of work that has found a solution to this problem. Unfortunately, some groups use qPCR regardless even when stable reference genes have not been validated, we decline to do so.

Regarding the upregulation of motility, this would definitely be an interesting line of inquiry for future work. We do speculate in the text that this phenotype could result from dysregulation of the cyclic di-GMP regulon (lines 203-219), and at this stage, it is impossible for us to predict if the phenotype is adaptive, and if so, if it benefits phage or host cells. The quick kinetics of ICP1 infection present a challenge to visualizing cell motility during infection, and at this time we have not developed a method for doing so.

Minor comments:

1. Is ICP1_2006 considered the 'type' strain? What was the clinical context / scientific rationale for picking this strain? Line 505

Because the Δ CRISPR mutant of ICP1_2006 cannot overcome any of the PLE variants, we are able to compare all of the PLEs' responses to this particular phage. This is not the only ICP1 isolate with that characteristic, but previous experimental characterization of PLE and ICP1 was predominately done using this strain (i.e. Refs 15, 24) so we opted to perform the experiments described with this phage isolate.

To clarify for the reader, we now explain this rationale in our materials and methods by saying "In continuity with previous PLE related studies(15, 24), ICP1_2006_E engineered to lack CRISPR-Cas (Δ CRISPR, Δ cas2-3) (23) was used for all experiments. (Lines 584 to 586).

2. Was there an attempt to repeat the collection at a lower O.D. before quorum sensing triggers in order to differentiate density dependent transcriptional events? Consider adding to the discussion.

We did not try to investigate the effects of quorum sensing on our system. While it is possible that there may be some effect by quorum sensing on PLE or ICP1 activity, all infections were done at the same OD, growth phase, and culture conditions, so we are confident that the differences we do report are due to PLE or ICP1.

3. Was induction of ctx phage observed in AKI media (1.5% Bacto-Peptone [Difco], 0.4% yeast extract, 0.5% NaCl). If no, why not. Line 507.

We recognize that AKI conditions are an established method for inducing TCP production, but we opted to use a ToxT expression construct due to the relative ease of constructing and working with this strain under our typical growth conditions. We address this in materials and methods under the heading 'CTX transduction assays'.

4. Add company name/ core lab that made rabbit anti ICP1 capsid. Line 597.

The company name is Genscript as indicated in the text (now line 677).

5. The estimate that that 25% of reads at 4 minutes are ICP1 transcripts is impressive (line 166). With respect to VC, why is there a lack of balance between the reads from the two VC chromosomes?

Relative to chromosome length, more essential growth functions are encoded on the large chromosome than the small chromosome, and the small chromosome has many small hypothetical genes of unknown function. The bias towards expression of large chromosome genes has been previously recognized (refs 30 and 31). Many small chromosome genes are upregulated when *V. cholerae* is growing within a host, suggesting that they may have a role in *in vivo* growth (ref 31).

To highlight that our observations are in line with previous work with have added the lines “As had been noted previously, the *V. cholerae* transcriptome skews heavily towards genes on the large chromosome (30, 31)” (line 167-169)

September 24, 2020

Dr. Kimberley D Seed
University of California, Berkeley
Department of Plant & Microbial Biology
271 Koshland Hall
Berkeley, CA 94720

Re: mSystems00358-20R1 (A family of viral satellites manipulates invading virus gene expression and can affect cholera toxin mobilization)

Dear Dr. Kimberley D Seed:

Your manuscript has been editorially accepted, and I am forwarding it to the ASM Journals Department for publication. For your reference, ASM Journals' address is given below. Before it can be scheduled for publication, your manuscript will be checked by the mSystems senior production editor, Ellie Ghatineh, to make sure that all elements meet the technical requirements for publication. She will contact you if anything needs to be revised before copyediting and production can begin. Otherwise, you will be notified when your proofs are ready to be viewed.

Sincerely,

Seth Bordenstein
Editor, mSystems

Journals Department
Tables S1, Tabs 1-5: Accept

Fig. S6: Accept

Fig. S5: Accept

Fig. S4: Accept

Fig. S1: Accept

Fig. S7: Accept

Fig. S8: Accept

Fig. S3: Accept

S1 Data: Accept

Fig. S2: Accept